# Childhood B-Cell Preleukemia Mouse Modeling

**DOI:** 10.3390/ijms23147562

**Published:** 2022-07-08

**Authors:** Marta Isidro-Hernández, Silvia Alemán-Arteaga, Ana Casado-García, Belén Ruiz-Corzo, Susana Riesco, Pablo Prieto-Matos, Jorge Martínez-Cano, Lucía Sánchez, César Cobaleda, Isidro Sánchez-García, Carolina Vicente-Dueñas

**Affiliations:** 1Experimental Therapeutics and Translational Oncology Program, Instituto de Biología Molecular y Celular del Cáncer, CSIC-USAL, Campus M. de Unamuno s/n, 37007 Salamanca, Spain; martaisidroh@usal.es (M.I.-H.); alemanarteaga.silvia@usal.es (S.A.-A.); anacasagar@usal.es (A.C.-G.); belenruizcorzo@gmail.com (B.R.-C.); isg@usal.es (I.S.-G.); 2Institute for Biomedical Research of Salamanca (IBSAL), 37007 Salamanca, Spain; sriesco@saludcastillayleon.es (S.R.); pprieto@saludcastillayleon.es (P.P.-M.); 3Department of Pediatrics, Hospital Universitario de Salamanca, Paseo de San Vicente, 58-182, 37007 Salamanca, Spain; 4Immune System Development and Function Unit, Centro de Biología Molecular Severo Ochoa (Consejo Superior de Investigaciones Científicas-Universidad Autónoma de Madrid), 28049 Madrid, Spain; jorge.martinez@cbm.csic.es (J.M.-C.); cesar.cobaleda@csic.es (C.C.); 5School of Law, University of Salamanca, 37007 Salamanca, Spain; id00698454@usal.es

**Keywords:** preleukemic cells, childhood leukemia, leukemia, mouse modeling, genetic predisposition

## Abstract

Leukemia is the most usual childhood cancer, and B-cell acute lymphoblastic leukemia (B-ALL) is its most common presentation. It has been proposed that pediatric leukemogenesis occurs through a “multi-step” or “multi-hit” mechanism that includes both in utero and postnatal steps. Many childhood leukemia-initiating events, such as chromosomal translocations, originate in utero, and studies so far suggest that these “first-hits” occur at a far higher frequency than the incidence of childhood leukemia itself. The reason why only a small percentage of the children born with such preleukemic “hits” will develop full-blown leukemia is still a mystery. In order to better understand childhood leukemia, mouse modeling is essential, but only if the multistage process of leukemia can be recapitulated in the model. Therefore, mouse models naturally reproducing the “multi-step” process of childhood B-ALL will be essential to identify environmental or other factors that are directly linked to increased risk of disease.

## 1. Introduction

Leukemia is among the main causes of disease-related childhood mortality, and B-cell acute lymphoblastic leukemia (B-ALL) is the most common form of the disease [1,2]. Nowadays, the current long-term survival or cure rate is very high at ~90%, but treatment is traumatic, toxic, and associated with long-term health consequences [3]. Before we can offer new therapeutic approaches to children or, better yet, establish preventive measures to avoid the development of leukemia, it is necessary to know the biological mechanisms of the disease. Although the cause or causes of acute leukemias are not known with accuracy, it is well known that there are several factors that predispose someone to suffer from these hemopathies. These factors include genetics, immunodeficiencies, certain environmental factors and pathogens. Regarding the first, several facts support the participation of genetic factors in the development of acute leukemias [4,5,6]. Even though the process of leukemogenesis has not been completely clarified, in recent years there have been great advances towards the understanding of the mechanisms that determine the malignant transformation of hematopoietic precursors [6,7]. However, even so, much remains to be understood since this knowledge has not been transformed into a direct benefit for the patient.

## 2. The Multistep Pattern of Childhood B-ALL

With regard to B-ALL, the identification of specific chromosomal alterations and the advances in molecular biology techniques have made it possible to discover that a fundamental mechanism of leukemogenesis is the alteration of proto-oncogenes [6,7,8]. A “two-step” model has been proposed to explain the development of the most common forms of B-ALL [9]; in this model, the first step is the existence of a genetic alteration, either acquired in utero or inherited as a germline genetic variant (Figure 1). Systematic screens aimed at determining the frequency of the first-step mutation found that a significant percentage (>5%) of healthy newborns carry preleukemic clones [10,11], therefore suggesting that the appearance of preleukemic lesions in utero is far more common than indicated by the incidence of disease (as less than 1% of genetic carriers will develop the disease), and evidencing the fact that a second oncogenic step is required.

Further support for a “multi-hit” mechanism of B-ALL development comes from the relatively limited concordance of B-ALL in monozygotic twins, and from differences in the time course of leukemia development in monozygotic twins carrying identical genetic alterations [12,13,14]. It has been proposed that the preleukemic cell, originated by a leukemia-initiating mutation, is shared in utero between both twins due to intraplacental metastasis [13]. However, such preleukemic clones can reside in healthy twins, in such a way that only one twin will develop B-ALL (the one in which a secondary, post-natal, event takes place) but not the healthy twin [15,16,17,18]. These preleukemic cells can persist for years, without harm to the child, but, when there is an exposure to an oncogenic environment, this provides the necessary selective pressure for the acquisition of secondary genomic alterations (second-step) and for the appearance of leukemia (Figure 1). However, the precise nature of these oncogenic environments is still unclear; cooperating oncogenic mutations will appear perinatally or during infancy, likely as consequence to the exposure to infectious agents that are challenging an already dysregulated, preleukemic, immune system [19,20,21]. This model (known as “the infective theory”) was the first suggested causal exposure for childhood B-ALL and remains the one with the strongest supporting pieces of evidence [9,21,22,23]. There are also other epidemiologic risk factors that are preferentially associated with certain cytogenetics subtypes of ALL, as is the case for hyperdiploid, and ETV6-RUNX1 subtypes that are connected with home paint exposure [24]. This highlights the fact that the diverse subtypes of this disease might have a different etiology. What is clear is that both steps are essential for overt leukemia development, and that preleukemic cells could be around in healthy individuals who will never develop leukemia (Figure 1). Why these abnormal cells do not progress to fully transformed cells is a key question that needs to be solved, and for which the capacity of modeling these preleukemic stages in the mouse will be essential.

## 3. Mouse Modeling of Genetic Susceptibility to Childhood Leukemia

Although the genomic landscape of patients with childhood leukemia has been extensively characterized [4,5,6], and extensive epidemiologic research has been carried out to try to associate different risk factors with B-ALL development, the extrinsic factors that promote the conversion of the preleukemic clone into full-blown B-ALL are not yet understood. To fully understand the etiology of pediatric leukemia, and how the interactions between a given genotype and certain environmental exposures (gene–environment interactions) may cause the appearance of the disease, we need animal models that resemble childhood leukemia evolution.

In vivo research in mouse models integrates all the organization levels relevant for the development of the disease, from the complexity of the organs and their different cell types to the global physiological status of the organism. Animal models have greatly contributed to our understanding of the natural history of B-ALL from the preleukemic state to the fully transformed stages. For this reason, mouse models mimicking the genetic susceptibility to childhood leukemia can be instrumental to deciphering the associations between leukemia and environmental/epidemiological factors and to understanding why there are healthy carriers of the preleukemic cells that will never develop the disease.

The previously mentioned “two-step” model for B-ALL development should be used as the pivotal reference to model the disease in mice. Accordingly, the model should display the first hit that is either acquired in utero (prenatally acquired somatic mutations) or as a constitutional germline genetic variant (Figure 1). The second hit could be also modeled in the mice but it is much more valuable to use mouse models harboring just the first oncogenic event to assess potential risk factors that promote the spontaneous acquisition of the secondary events.

Mouse models of childhood leukemia should exhibit both phenotypic and genotypic concordance with the human disease being modeled. To this end, the appropriate cell of origin in B-ALL should be identified and targeted with the same genetic alteration present in human leukemia. As we will present in the following section, germline genetic predisposition should be present in all the cells of the mouse model by definition, but in the case of the first and/or the second somatic mutations the target cell of origin should be selected with caution.

Several mouse models have been developed to date that reproduce different genetic susceptibilities to develop B-ALL (Table 1), carrying either germline or prenatally acquired somatic mutations, as discussed below.

### 3.1. Modeling Germline Susceptibility

#### 3.1.1. Modeling B-ALL Driven by PAX5 Deletions/Mutations

PAX5 is the master transcription factor controlling B cell identity. Around 30% of pediatric B-ALLs present somatic mutations or deletions affecting *PAX5* [7]. Furthermore, inherited mutations of *PAX5* have been recently identified defining a new syndrome of susceptibility to B-ALL, therefore extending the role of PAX5 alterations in the pathogenesis of B-ALL [25,26]. These inherited mutations seem to cause the appearance of a persistent, latent, preleukemic clone that might give rise to full-blown B-ALL in only 30% of the family members who carry the mutation [25]. Thus, this PAX5 B-ALL syndrome also shows incomplete penetrance and, also in this case, the mechanisms responsible for the evolution from the preleukemic condition into full-blown B-ALL have been unknown until the development of mouse models mimicking this scenario. *Pax5^+/−^* mice were generated in 1994 [27], and they were used to study B-ALL etiology in the context of *PAX5* familial mutations in 2015 [19]. In this study, it has been shown that postnatal exposure to common pathogens promotes the development to B-ALL in *Pax5* heterozygous mice. These murine B-ALLs closely resemble the human disease in terms of phenotype, incidence and also in the nature of secondary mutations.

Infections have always been a usual suspect taken into account to explain the development of childhood leukemia [22], but this possibility had not been proven experimentally until *Pax5* heterozygous mice were exposed to common infections. The penetrance/incidence of the disease in exposed *Pax5^+/−^* mice is 22% [19], even though all the mice are genetically identical and all of them are exposed to the same infectious agents; therefore, there must be a factor that is modulating the development of the disease at the level of the individual, closely resembling human disease. The mice that develop leukemia also exhibit a distinct gut microbiome that could potentially be used as a biomarker to identify B-ALL susceptibility [28]. Remarkably, altering the gut microbiome by treating the mice with antibiotics is sufficient to trigger B-ALL development, even in the absence of infectious stimuli [28], thus highlighting the importance of the gut microbiota as a barrier for B-ALL in the context of a *Pax5* genetic susceptibility.

It has been suggested that infection or chronic inflammation triggers B-ALL through the induction of activation-induced cytidine deaminase (AID, also known as AICDA) expression in preleukemic cells [9,29]. However, using the above-mentioned *Pax5* heterozygous mice, it has been proven that AID does not impact either the latency or the incidence of infection-mediated *Pax5^+/−^* B-ALL development [30]. This model has also been useful to show that those leukemias are independent of T-cells [28]. Therefore, the *Pax5^+/−^* model has been instrumental to deciphering the associations between genetic predisposition to leukemia and environmental factors.

Moreover, chemically-*N*-ethyl-*N*-nitrosourea (ENU) and retrovirally mediated Moloney murine leukemia retrovirus (MMLV) mutagenesis in *Pax5^+/−^* mice resulted in a significantly increased penetrance of leukemia [31]. However, this finding has only been observed in previously surgically thymectomized mice that do not fully reproduce the human scenario. However, on the other hand, genomic analysis of *Pax5^+/−^* mice subjected to mutagenesis identified secondary genetic alterations that recapitulate those observed in human leukemia, as is the case of secondary alterations of *Pax5* or recurrent mutations in genes such as *Jak1*, *Jak3*, *Ptpn11*, and *Nras* [31].

*Pax5* heterozygous mice have been also used to study the cooperation between *Pax5* mutations and the increased expression of STAT5, which is correlated with poor prognosis in ALL patients. The coexpression of *Pax5^+/−^* and *Stat5b-CA* (a constitutively active form of STAT5) induce ALL in 100% of the transgenic mice [32], indicating that Stat5 activation is the necessary second hit that synergize with Pax5 heterozygosity and fully transforms the preleukemic *Pax5^+/−^* cells.

A recent study has modelled in mice a single point mutation found in the patient’s *PAX5* gene, trying to better recapitulate the human preleukemic stage. In this report, aged *Pax5^Y351 */Y351 *^* mice spontaneously develop B-ALL with 100% penetrance [33].

Similar to *Pax5^+/−^* mice, *Irf4*^−/−^ mice also spontaneously develop B-ALL with incomplete penetrance (incidence 17.5%), when stressed with bacterial compounds (such as lipopolysaccharide (LPS), an important outer membrane component of gram-negative bacteria), due to the acquisition of *Jak3* mutations [34]. Although the transformation process in *Pax5^+/−^* B-ALL is independent of AID [30], in Irf4^−^^/−^ leukemias, AID might be playing a role in the acquisition of secondary mutations. Moreover, in this preleukemia mouse model, not all the healthy carriers develop the disease and support the “two-hit” leukemogenesis model where the loss of *Irf4* is the first hit and the subsequent Jak3 mutations act as the second hit.

In a much more relevant way, modeling *Pax5* deletions/mutations in mice has provided the first in vivo evidence for a potential preventive strategy for B-ALL development [35]. Casado-García et al. have shown that the elimination of the preleukemic cells early in life can significantly mitigate the risk of B-ALL. To this aim, *Pax5^+/−^* mice are transiently treated with ruxolitinib, a JAK1/2 inhibitor that preferentially targeted *Pax5^+/−^* versus wild-type B-cell progenitors and leads to B-ALL prevention [35]. Thus, childhood B-cell preleukemia mouse modeling will serve in the future to develop novel approaches to prevent B-ALL onset by selectively targeting the preleukemic clone.

#### 3.1.2. Modeling B-ALL Driven by IL7R Activating Mutation

Within the category of “Ph-like” B-ALLs [36], the majority are characterized by the aberrant expression of CRLF2/IL7RA and activating mutations of the JAK-STAT signaling pathway [36], such as IL7RA activating mutations [37]. The axis constituted by interleukin-7 (IL-7) and its receptor (IL-7R), composed in turn by IL-7Rα (encoded by IL7R) and γc (encoded by IL2RG) chains, is essential for normal lymphoid development [38]. *IL7R* activating mutations co-occur in B-ALL with loss-of-function mutations in the negative signalling regulator *SH2B3* or alterations in the transcription factor *IKZF1* [39].

Recent research has described how *IL7RA* activation can trigger B-cell precursor ALL in mouse hematopoietic cells with incomplete penetrance [40,41]. When those preleukemic cells (expressing the activating mutation of *IL7R*) also lose the expression of *Sh2b3* or *Ikzf1*, leukemia development exacerbates significantly [40], corroborating findings from human patients [39]. Furthermore, homozygous expression of mutant *IL7R* leads to early disease onset and full penetrance [41], suggesting that highest IL-7R signalling levels easily transform B cell precursors without requiring additional oncogenic hits.

Moreover, in human hematopoietic cells, the expression of an activated form of the IL7RA generates an initiating preleukemic state that is vulnerable to evolving to overt “Ph-like” B-ALL with a nearly complete penetrance [42]. Secondary mutations are necessary to mediate the progression of the preleukemic state to overt malignancy, and a selective pressure of serial transplantation can promote the evolution of preleukemic cells into leukemia [42]. Moreover, loss of the negative cell cycle regulator CDKN2A is commonly detected in human B-ALL with mutated activated *IL7RA* [4]. Therefore, the loss of CDKN2A may be the trigger for the evolution of the preleukemic cells carrying an *Il7R* activating mutation to a fully transformed leukemic stage. This hypothesis has been modelled and confirmed in mice, where an activated IL7R initiates a preleukemic state that evolves to “Ph-like” B-ALL through the loss of CDKN2A [42].

Therefore, with the help of the models developed so far, the mechanism by which *IL7R* activating mutations lead to B-ALL has been understood in more detail, and this knowledge could now be used for the development of more specific therapies.

### 3.2. Modeling Somatic Susceptibility

#### 3.2.1. Modeling B-ALL Driven by the ETV6-RUNX1 Fusion Gene

The *ETV6-RUNX1* fusion gene is the most frequent chromosomal alteration in pediatric cancer and occurs in approximately 25% of childhood B-ALLs: it defines a subgroup of patients with an excellent prognosis [43]. The t(12;21)(p13;q22) chromosomal translocation results in the fusion of two critical regulators of hematopoiesis, bringing together the 5′ portion of the *ETV6* (*TEL*) gene on chromosome 12p13 and nearly the entire *RUNX1* (*AML1*) gene on chromosome 21q22 [44].

Although the ETV6-RUNX1 oncogenic lesion can be frequently (>5%) found in otherwise healthy neonatal cord blood, only a few *ETV6-RUNX1* carriers will develop B-ALL, as the ETV6-RUNX1+ leukemia rate is around 0.01% [11]. These findings indicate that the *ETV6-RUNX1* gene fusion confers a low risk of developing B-ALL, and therefore constitutes the first hit in a process of leukemogenesis by giving rise to the appearance of a preleukemic clone, which will then require secondary postnatal genetic hits.

Since the *ETV6-RUNX1* fusion gene is very specifically associated with an ALL of B cell phenotype, it has been a paradigm in the field, and many genetically engineered mouse models have been generated over the last decades, designed to express the *ETV6-RUNX1* gene in a committed B-cell [45,46,47,48,49,50]. However, these models have systematically failed to develop leukemia. For this reason, the newer mouse models have focused their design not on B cells, but on targeting the expression of ETV6-RUNX1 to more immature, plastic, hematopoietic cells. With this approach, it has been shown that limiting ETV6-RUNX1 expression to murine stem cells can indeed induce childhood B-ALL development upon infection exposure [20]. In this mouse model, *ETV6-RUNX1* gene expression is restricted to hematopoietic stem/progenitor cells (HS/PCs) by taking advantage of the locus control region of the *Sca1* gene. These *Sca1-ETV6-RUNX1* mice developed exclusively B-ALL at a low disease penetrance, and only when they were exposed to common pathogens [20], hence again supporting the infective theory of childhood leukemia development [22]. The fact that limited expression of ETV6-RUNX1 during early hematopoietic development conferred only a low risk of developing B-ALL, and only after exposure to common pathogens (10% incidence), explains the low incidence of B-ALL in human preleukemic carriers, and further validates the accuracy of this model. The infection-driven B-ALLs arising in *Sca1*-*ETV6-RUNX1* mice not only recapitulate the human disease in its low penetrance, but also in its general pathology and its genomic lesions [20]. Therefore, ETV6-RUNX1 is necessary for the early stages of transformation, but the final tumor phenotype is determined by the second hit suffered by the preleukemic hematopoietic precursor [51]. With all these evidences, the *Sca1-ETV6-RUNX1* model closely mimics human *ETV6-RUNX1* preleukemic biology and therefore provides a means to evaluate the oncogenic potential of different environmental exposures that might play a role in B-ALL development, as is the case for common infections [19] or other environmental factors [52].

In a different mouse model, *ETV6-RUNX1* expression was driven from the endogenous *Etv6* promoter and can be targeted to specific hematopoietic populations in combination with the Cre-LoxP recombinase system [53]. When *ETV6-RUNX1* is expressed in hematopoietic stem cells (HSCs), but not in early lymphoid progenitors, these mice are prone to malignancies, but need a chemical mutagenesis (ENU) to transform the preleukemic cells [53]. Furthermore, the observed malignancies were of T-lymphoid lineage, hence not corresponding with the human ETV6-RUNX1 B-ALL scenario.

In a similar, but constitutive, knock-in approach, the endogenous mouse *Etv6* locus was targeted to express the *Etv6-RUNX1* fusion [54]. These mice do not develop B-ALL spontaneously. However, when an insertional mutagenesis screen was performed by inter-crossing these mice with those carrying a Sleeping Beauty transposon array, 20% of the offspring developed B-ALL, but even more often T-cell leukemia and myeloid leukemias appeared [54]. Therefore, also in this case, there is a discrepancy with human B-ALL, as the *ETV6-RUNX1* fusion gene is only associated to ALL of B phenotype in humans.

#### 3.2.2. Modeling B-ALL Driven by the E2A-PBX1 Fusion Gene

The t(1;19) chromosomal translocation fuses the transcription factor E2A (TCF3) with the homeobox gene PBX1, and constitutes the first hit in a specific subtype of B-ALL, which is present in 5–7% of childhood ALLs [55,56]. As in the case of the ETV6-RUNX1 fusion, the frequency of newborns carrying the t(1;19) chromosomal translocation (≈0.6%) exceeds the corresponding E2A-PBX1 + leukemia incidence (∼0.002%) [10]. Although patients with E2A-PBX1 ALL have a favourable 5-year prognosis with a ∼90% survival, the disease can also have a more aggressive course and is associated with an increased risk of central nervous system relapse [57]. This, together with the strong secondary effects of the toxic chemotherapy used for its treatment, motivate the search for its molecular mechanisms and potential therapeutic targets.

Modelling this subtype of B-ALL in mice has shown that animals conditionally expressing the *E2A-PBX1* fusion oncogene develop B-ALL with a varied incidence from 5% to 50%, depending on the Cre recombinase (*Cd19*, *Mb1*, or *Mx1*) used to induce *E2A-PBX1* expression [58]. This mouse model has been really useful to bring to light the “multi-step” pathogenesis associated with E2A-PBX1-driven B-ALL. In preleukemic mice, conditional *Pax5* deletion cooperated with E2A-PBX1, leading to an increased leukemia penetrance and shortening its latency [58], hence confirming a tumor-suppressive role for *Pax5* in an E2A-PBX1 background. This cooperation between the loss of *Pax5* and the driver oncogenic fusion gene is similar to that previously observed for ETV6-RUNX1 [51] and BCR-ABL [59] in the genesis of B-ALL. Moreover, modelling E2A-PBX1-induced B-cell leukemia in T-cell deficient *CD3ε*^−/−^ mice [60] showed that the *Hoxa* gene cluster is preferentially targeted in E2A-PBX1-induced tumors, thus suggesting a functional collaboration between these genes in pre-B-cell tumors.

#### 3.2.3. Modeling B-ALL Driven by the *BCR-ABL* Fusion Gene

The t(9;22)(q34;q11) translocation, generating the *BCR-ABL1* gene, coding for a constitutively active tyrosine kinase, is present in 3–5% of pediatric B-ALL and in 25% of adult B-ALL patients [5].

Preleukemic clones carrying the *BCR-ABLp190* oncogenic lesions are often found in neonatal cord blood [17]. Classic mouse models of transgenic-driven BCR-ABL ALL showed a 100% penetrance of B-ALL disease [61,62], something that does not happen in human patients, therefore not allowing one to explore the preleukemic transformation process similarly taking place. More recently, a more faithful model has been generated in transgenic mice where the *BCR-ABL^p190^* expression is restricted to HS/PC: the *Sca1-BCR-ABLp190* mice. In these mice, B-ALL again has a low penetrance, which resembles the human disease [59]. In the *Sca1-BCR-ABLp190* model, the restriction of *BCR-ABL**^p^*^190^ to HS/PC also primed the cells to a preleukemic stage (first hit) and afterward this genetic insult is not required anymore. Interestingly, introducing the second hit (*Pax5* deletion) from the beginning, together with the appearance of the preleukemic clone, induces leukemic transformation in all the animals [59]. This indicates that the first hit (*BCR-ABL**^p^*^190^) is priming the cells to a preleukemic state, but it is the second hit (*Pax5* deletion) acting as the real driver of B-ALL development. In a similar way, the loss of the tumor suppressor gene *IKZF1* (IKAROS) has been shown to cooperate with BCR-ABL in a transgenic model of ALL [63].

It is of note that current mouse BCR-ABL1-driven models do not discriminate between childhood and adult B-ALL cases. Thus, further investigation is required to understand the different frequencies of this subtype of leukemia in pediatric and adult patients.

#### 3.2.4. Modeling B-ALL Driven by the PAX5 Translocation

We have seen the role played in leukemia development by inherited mutations affecting the Pax5 locus. Besides these congenital alterations, also de novo translocations affecting PAX5 appear at a frequency of 2.5% in human B-ALLs. These translocations are very diverse and are currently known to involve 28 different partner genes, generating novel PAX5 fusion proteins [4]. The different partner genes code for proteins of diverse functions such as transcription factors (ETV6), signal transducers (JAK2), chromatin regulators (BRD1) and structural proteins (ELN), among others [64].

Some of these *PAX5* translocations found in humans have been modelled in mice to try to unravel the role of these fusion genes. Thus, it has been shown that PAX5-ETV6 functions as a potent oncoprotein to promote B-ALL development in combination with loss of the tumor suppressors Cdkn2a and Cdkn2b [65]. In a similar way, in a knock-in mouse model in which the *PAX5-ELN* transgene is expressed specifically in B cells, 80% of the mice developed B-ALL [66]; in these animals, leukemic transformation needs recurrent secondary mutations on genes affecting key signalling pathways required for cell proliferation [66]. Therefore, it seems that PAX5-ETV6 and PAX5-ELN translocations, similar to heterozygous inherited *PAX5* alterations [19], promote B-ALL development in cooperation with a second oncogenic “hit”.

Conversely, *Pax5^Jak2/+^* mice (a mouse model of the PAX5-JAK2 fusion protein) rapidly developed an aggressive B-ALL in the absence of cooperating mutations [67]. In this scenario, the PAX5-JAK2 oncogene (first hit) leads to STAT5 activation by maintaining phosphorylated STAT5 levels in the nucleus (second hit) [67]. Thus, the PAX5-JAK2 fusion protein functions as a dual-hit mutation to promote B-ALL with a short latency and high penetrance, and does not require a cooperating second event for the full transformation of the preleukemic clone.

**Table 1 ijms-23-07562-t001:** Mouse models of childhood B acute lymphoblastic leukemia.

	Genetic Susceptibility	Transgene	Manipulation	Penetrance	Phenotype	Ref.
** Modeling Germline susceptibility **	*Pax5* ^+/−^	*Pax5* ^+/−^	Exposed to common infections or gut microbiome dysbiosis	22%	B-ALL	[19,28,30]
	*Pax5* ^+/−^	*Pax5* ^+/−^	Thymectomy plus ENU or MMLV	100%	B-ALL	[31]
	*Pax5* ^+/−^	*Pax5+/− X Stat5b-CA*	None	100%	B-ALL	[32]
	*PAX5^Y351*/+^*	*PAX5^Y351 */ Y351 *^*	None	100%	B-ALL	[33]
	*Irf4* ^−/−^	*Irf4* ^−/−^	None	~17%	B-ALL	[34]
	*IL7R* activating mutation	*Rosa26-aIL7R+/wt* (Mb1 Cre and CD19-driving Rosa26 promoter)	None	~20%	B-ALL	[40]
	*IL7R* activating mutation	*IL7R^cpt/wt^* (CD2 Cre-driving LoxP FLEX switch)	None	>60%	B-ALL	[41]
	*IL7R* activating mutation	*IL7Rains* (Eμ-B29 promoter/enhancer)	Transduced in human CD34+ cells and transplanted into NOD/LtSz-scid IL2Rγnull (NSG) mice	100%	B-ALL	[42]
** Modeling Somatic susceptibility **	*ETV6-RUNX1*	*ETV6-RUNX1 (Sca1 promoter)*	Exposed to common infections	10%	B-ALL	[20]
	*ETV6-RUNX1*	*ETV6-RUNX1 (Mx Cre-driving Etv6 promoter)*	chemical mutagenesis /ENU)	<30%	T-cell malignancy	[53]
	*ETV6-RUNX1*	*ETV6-RUNX1*	Retroviral gene transfer	~22%	B and T-ALL	[68]
	*ETV6-RUNX1*	*Etv6^+/RUNX1-SB^*	Sleeping beauty (SB) transposon system	3%	B, T and myeloid-ALL	[54]
** Modeling Somatic susceptibility **	*ETV6-RUNX1*	*Etv6^+/RUNX1-SB^; Pax5^+/^* ^−^	Sleeping beauty (SB) transposon system	26%	B, T and myeloid-ALL	[69]
	*ETV6-RUNX1*	*ETV6-RUNX1 (Sca1 Cre-driving Etv6 promoter)*	Exposed to common infections	6–34%	B and T-ALL	[51]
	*E2A-PBX1*	*Tg E2A-PBX1 (Cd19 Cre, Mb1 Cre, or Mx1 Cre-driving E2A promoter)*	None	5–50%	B-ALL	[58]
	*E2A-PBX1*	*E2a–PBX1 (TCR V and Lck promoters) × CD3ε* ^−/−^	None	13–40%	B and T-ALL	[60]
	*BCR-ABL*	*BCR-ABL^p190^ (Sca1 promoter); Pax5^+/^* ^−^	None	90%	B-ALL	[59]
	*BCR-ABL*	*BCR-ABL^p190+/0^ (metallothionein promoter)*	None	80%	B and myeloid-ALL	[61]
	*BCR-ABL*	*bcr-ABL/bcr+*	None	~100%	B-ALL	[62]
	*BCR-ABL*	*BCR-ABL^p190+/0^ (metallothionein promoter); IK^L/+^*	hypomorphic IKZF1 allele	100%	B-ALL	[63]
	*PAX5-JAK2*	*PAX5-JAK2*	None	100%	B-ALL	[67]
	*PAX5-ELN*	*PAX5-ELN* (IgH promoter)	None	80%	B-ALL	[66]
	*PAX5-ETV6*	*PAX5 ^ETV6/+^* (Pax5 promoter)	Cdkna2a/b^+/−^	98%	B-ALL	[65]

Note: ENU: N-ethyl-N-nitrosourea; MMLV: retrovirally mediated Moloney murine leukemia retro-virus; B-ALL: B-cell acute lymphoblastic leukemia; T-ALL: T-cell acute lymphoblastic leukemia.

## 4. Concluding Remarks

In order to better understand childhood leukemia, mouse modeling is instrumental, but models will only be really useful if the “multi-stage” process of leukemia, accepted for the vast majority of B-ALL subtypes, is naturally recapitulated. A good animal model should also demonstrate both phenotypic and genotypic concordance with the human disease being modelled. To this end, the suitable cell of origin in B-ALL should be identified and targeted with the appropriate genetic alteration. In the recent years, it has been suggested that the first genetic alteration in a non-committed cell has the capacity to induce a genetic priming in the cell-of-origin that impose the ultimate phenotype of the transformed cell [70]. Since childhood B-ALL is a heterogeneous disease, several mouse models recapitulating the different genomic subtypes should be developed in order to characterize the different molecular and cellular subtypes of the disease.

The models reviewed in this article constitute powerful tools to characterize new molecular and cellular players in B-ALL aetiology, in the context of normal hematopoietic development. These models should allow us to test new therapeutic approaches in an immunocompetent environment. The current mouse models developed so far, and the ones that will be developed in the near future, should facilitate the identification of the genetic basis of the evolution from the preleukemic clone to full-blown B-ALL, and should help us to understand why these abnormal preleukemic cells do not progress to fully transformed cells in a great number of healthy carriers. Those models will also serve as tools to develop specific approaches to target the preleukemic B-cells as a means to prevent B-ALL development in the future.

## Figures and Tables

**Figure 1 ijms-23-07562-f001:**
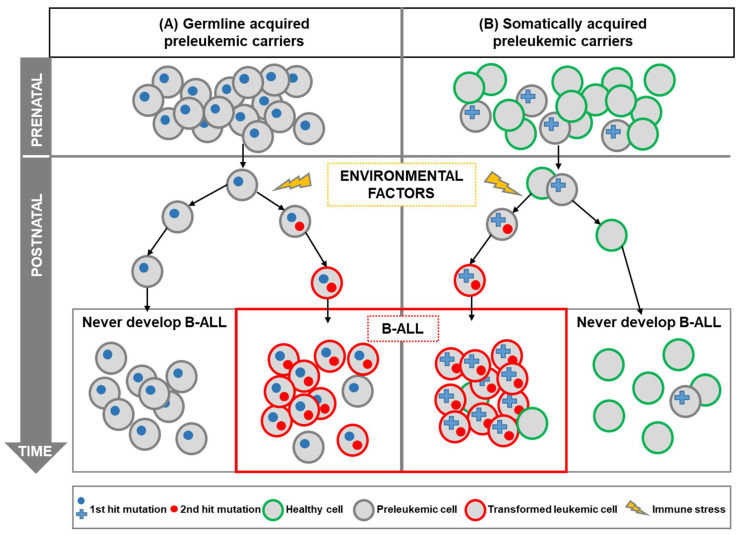
**Childhood B-ALL development in genetically predisposed carriers.** A “two-hit” model has been proposed for most cases of B-ALL. B-ALL development requires an initiating mutation acquired prenatally (first hit), which leads to the preleukemic cell generation, as well as a second postnatal mutation (second hit). Genetic susceptibility to childhood leukemia (first hit) could be due to inherited genetics (**A**) or due to prenatal somatic aberrations such as chromosomal aberrations (**B**). In both cases, the genetic alteration gives rise to preleukemic cells that are susceptible to transformation. In this model, the second hit is triggered by postnatal environmental factors. Recent pieces of evidence have shown that those environmental factors, such as infection, can promote immune stress in the preleukemic cells leading to the development of overt leukemia. It is of note that the preleukemic cells could be around in healthy individuals who will never develop leukemia, as the incidence of childhood B-ALL is much less frequent than the frequency of preleukemic healthy carriers.

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
