# Peer review of "Childhood B-Cell Preleukemia Mouse Modeling"

_ijms, 2022, doi:10.3390/ijms23147562_

Round 1

Reviewer 1 Report

The authors have written a comprehensive review on murine models of B-cell leukemia.   

Could they include the frequency that the in utero and subsequent post-natal detection of mutations.   Is for example somatic mutations more likely to develop leukemia than germline.  This could be infered from Figure 1 as 2 out 3 cells acquire 2nd mutations as opposed to 2 out of 4 for germline? 

Some frequency of the mutations might be useful so that the reader can see if one specific mutation is more studied than others. 

Some discussion on the methods used to introduce 1st and/or 2nd mutations might be of value - another figure perhaps? - particularly for the germline acquired 1st mutations.  

In Table 1, the left hand column heading is forced on to the top cell of the Modelling somatic mutations - i guess this is a table formating issue - but it would be better if could be across all relevant rows.  

Could section 3 be split in to somatic and germline sections each containing the relevant genetic mutations?  

Author Response

Response to Reviewer 1 Comments

Reviewer 1

The authors have written a comprehensive review on murine models of B-cell leukemia.   

We want to thank the reviewer for his/her thorough review of our manuscript and his/her very positive comments and suggestions for improving our manuscript. The referee indicates, quite rightly, some weak points. Now we have carefully addressed all comments in the revised manuscript (all changes are marked up using the “Track Changes” function) as detailed below.

Point: Could they include the frequency that the in utero and subsequent post-natal detection of mutations.  Is for example somatic mutations more likely to develop leukemia than germline.  This could be infered from Figure 1 as 2 out 3 cells acquire 2nd mutations as opposed to 2 out of 4 for germline?

Some frequency of the mutations might be useful so that the reader can see if one specific mutation is more studied than others. 

Response: Thank you for this constructive comment. To address this, although there is no data for all the mutations linked to B-ALL, we have included the frequency of mutations in healthy newborns and the frequency of leukemia rate associated with these particular mutations. This is the case for ETV6-RUNX1 fusion (5% ETV6-RUNX1+ for newborns compared with 0.01% ETV6-RUNX1+ leukemia rate), and E2A-PBX1 fusion (0.6 % E2A-PBX1+ for newborns compared with ∼0.002% TCF3-PBX1+ leukemia rate) (please, see pages 8 and 9). For germline mutations, in the case of the Pax5 mutations, only 30% of the carriers will give rise to full-blown B-ALL (this point was already included in the manuscript, see page 6) but an extensive screening in newborns has not been done yet. There are other germline mutations linked to childhood ALL cases, for example, ETV6 mutations with also an incomplete penetrance of the disease. As it is a novel leukemia predisposition syndrome there are no mouse models available yet, and for this reason, have not been discussed in the current review. In a similar way, activating mutations in the IL7RA as the initiation of the preleukemic state is a novel finding and the is no data from newborns.

Point: Some discussion on the methods used to introduce 1st and/or 2nd mutations might be of value - another figure perhaps? - particularly for the germline acquired 1st mutations.  

Response: Thank you for this helpful comment. We have now discussed this issue in the revised version of the manuscript Sections 3 and 4 as follows.

Section 3: “Mouse models of childhood leukemia should exhibit both phenotypic and genotypic concordance with the human disease being modeled. To this end, the appropriate cell of origin in B-ALL should be identified and targeted with the same genetic alteration present in human leukemia. As we will present in the following section, germline genetic predisposition should be present in all the cells of the mouse model by definition, but in the case of the first and/or the second somatic mutations the target cell of origin should be selected with caution.” (please see page 3)

Section 4: “In recent years, it has been suggested that the first genetic alteration in a non-committed cell has the capacity to induce a genetic priming in the cell-of-origin that imposes the ultimate phenotype of the transformed cell [71].” (please see page 10)

Point: In Table 1, the left hand column heading is forced on to the top cell of the Modelling somatic mutations - i guess this is a table formating issue - but it would be better if could be across all relevant rows.  

Response: Thanks for noticing the mistake. We have fixed this issue in the revised file of the manuscript (please see page 5).

Point: Could section 3 be split into somatic and germline sections each containing the relevant genetic mutations?  

Response: Thank you for this constructive comment. We have split section 3 into “3.1. Modeling Germline susceptibility and 3.2. Modeling Somatic susceptibility” sections. In this way, now it fits better with the content of Table 1.

Reviewer 2 Report

The manuscript titled "Childhood B-cell preleukemia mouse modeling" by Isidro-Hernandez review the current mouse models that are used to recapitulate the multi-step processes leading to the onset of childhood B-ALL.

Overall, the manuscript is articulated and comprehensive. I have just a single minor comment concerning the acronyms (e.g. ENU) that should be indicated in-extenso when used for the first time in the manuscript.

Author Response

Response to Reviewer 2 Comments

Reviewer 2

The manuscript titled "Childhood B-cell preleukemia mouse modeling" by Isidro-Hernandez review the current mouse models that are used to recapitulate the multi-step processes leading to the onset of childhood B-ALL.

We would like to thank the reviewer for his/her thoughtful assessment of our manuscript and the constructive comment for improving it. Now we have carefully addressed all comments in the revised manuscript (all changes are marked up using the “Track Changes” function) as detailed below.

Point: Overall, the manuscript is articulated and comprehensive. I have just a single minor comment concerning the acronyms (e.g. ENU) that should be indicated in-extenso when used for the first time in the manuscript.

Response: Thank you for this constructive comment. To address this, we have described every acronym the first time is used in the text. We have highlighted all changes in the manuscript file using the “Track Changes” function (please see pages 5, 6, and 7).

Reviewer 3 Report

In the current manuscript, the authors summarized the mice models for B-cell acute lymphoblastic leukemia. The manuscript is well written and the contents are so informative for the readers.

1.     It would be helpful for the readers to understand the utility of the mice models if the authors could provide the penetrance of genetic susceptibility in human (e.g. how often do the cases with ETV6-RUNX1 develop B-ALL in human?)

2.     In the part of BCR-ABL1-driven model, the authors summarized the molecular mechanisms well, but it is still unclear why BCR-ABL1 is frequently found in adult cases. Discussion about this finding from the standpoint of mice model will be helpful for the readers to understand the pathophysiology of BCR-ABL1.

Author Response

Response to Reviewer 3 Comments

Reviewer 3

In the current manuscript, the authors summarized the mice models for B-cell acute lymphoblastic leukemia. The manuscript is well written and the contents are so informative for the readers.

We would like to thank the reviewer for his/her thoughtful assessment of our manuscript and the constructive comment for improving it. Now we have carefully addressed all comments in the revised manuscript (all changes are marked up using the “Track Changes” function) as detailed below.

  1. It would be helpful for the readers to understand the utility of the mice models if the authors could provide the penetrance of genetic susceptibility in human (e.g. how often do the cases with ETV6-RUNX1 develop B-ALL in human?)

Response: Thank you for this constructive comment. To address this, although there is no data for all the mutations linked to B-ALL, we have included the frequency of mutations in healthy newborns and the frequency of leukemia rate associated with these particular mutations. This is the case for ETV6-RUNX1 fusion (5% ETV6-RUNX1+ for newborns compared with 0.01% ETV6-RUNX1+ leukemia rate), and E2A-PBX1 fusion (0.6 % E2A-PBX1+ for newborns compared with ∼0.002% TCF3-PBX1+ leukemia rate) (please, see pages 8 and 9). For germline mutations, in the case of the Pax5 mutations, only 30% of the carriers will give rise to full-blown B-ALL (this point was already included in the manuscript, see page 6) but an extensive screening in newborns has not been done yet.

  1. In the part of BCR-ABL1-driven model, the authors summarized the molecular mechanisms well, but it is still unclear why BCR-ABL1 is frequently found in adult cases. Discussion about this finding from the standpoint of mice model will be helpful for the readers to understand the pathophysiology of BCR-ABL1.

Response: We agree with the referee that this question is reasonable and of interest. In the revised version of the manuscript, we have discussed this point as follows: “It is of note, that current mouse BCR-ABL1-driven models do not discriminate between childhood and adult B-ALL cases. Thus, further investigation is required to understand the different frequencies of this subtype of leukemia in pediatric and adult patients.” (please see page 9)